# Impact of Exposomes on Ocular Surface Diseases

**DOI:** 10.3390/ijms241411273

**Published:** 2023-07-10

**Authors:** Merrelynn Hong, Louis Tong, Jodhbir S. Mehta, Hon Shing Ong

**Affiliations:** 1Corneal and External Diseases Department, Singapore National Eye Centre, Singapore 168751, Singapore; merrelynn.hong@mohh.com.sg (M.H.); louis.tong.h.t@singhealth.com.sg (L.T.); jodhbir.s.mehta@singhealth.com.sg (J.S.M.); 2Ocular Surface Group, Singapore Eye Research Institute, Singapore 169856, Singapore; 3Department of Ophthalmology and Visual Science, Duke-NUS Medical School, Singapore 169857, Singapore; 4Tissue Engineering and Cell Therapy Group, Singapore Eye Research Institute, Singapore 169856, Singapore

**Keywords:** ocular surface, microbiome, microbiota, metagenomics, next-generation sequencing, diversity, cornea, conjunctiva, dry eye disease, contact lens, meibomian gland dysfunction, blepharitis, allergic eye disease, allergic conjunctivitis, vernal keratoconjunctivitis, cicatrising conjunctivitis, Stevens-Johnson syndrome

## Abstract

Ocular surface diseases (OSDs) are significant causes of ocular morbidity, and are often associated with chronic inflammation, redness, irritation, discomfort, and pain. In severe OSDs, loss of vision can result from ocular surface failure, characterised by limbal stem cell deficiencies, corneal vascularisation, corneal opacification, and surface keratinisation. External and internal exposomes are measures of environmental factors that individuals are exposed to, and have been increasingly studied for their impact on ocular surface diseases. External exposomes consist of external environmental factors such as dust, pollution, and stress; internal exposomes consist of the surface microbiome, gut microflora, and oxidative stress. Concerning internal exposomes, alterations in the commensal ocular surface microbiome of patients with OSDs are increasingly reported due to advancements in metagenomics using next-generation sequencing. Changes in the microbiome may be a consequence of the underlying disease processes or may have a role in the pathogenesis of OSDs. Understanding the changes in the ocular surface microbiome and the impact of various other exposomes may also help to establish the causative factors underlying ocular surface inflammation and scarring, the hallmarks of OSDs. This review provides a summary of the current evidence on exposomes in various OSDs.

## 1. Introduction

The ‘ocular surface’ is a complex integrated system which includes the corneal epithelium, the conjunctiva, the tear film, components of the eyelids (incorporating eyelashes and meibomian glands), the lacrimal gland, and the nasolacrimal duct [1]. A healthy ocular surface is important for preserving the transparency of the ocular media and comfort. Being the exposed outermost layer, the integrity of the ocular surface is crucial for protecting the eye against an adverse environment. All of the components of the ocular surface system are linked functionally by the continuity of its epithelium and work as an immunological unit which is capable of responding to external insults.

When ocular surface defences are breached by external insults such as infectious pathogens, autoimmunity, or trauma, a highly orchestrated innate immune response is triggered, bringing about acute inflammation [2]. Although it is designed to limit tissue injury and promote repair, the inflammatory response at the ocular surface is a double-edged sword. In many ocular surface diseases (OSDs), excessive and persistent inflammation result in damage to healthy by-standing tissues. Such uncontrolled inflammatory damage plays an important role in the pathophysiology of many OSDs, leading to significant ocular morbidity, including visual loss [3]. Nevertheless, due to the complexity of pathophysiological interactions and the lack of animal models, the processes that propagate inflammation and tissue damage in OSDs are poorly understood.

Exposomes refer to the totality of environmental factors that individuals are exposed to, and have been increasingly studied for their impact on ocular surface diseases. An exposome consists, broadly, of both external and internal environmental factors [4,5]. External factors may include environmental conditions such as the climate, urban or rural areas of living, social capital, stress, or more specific conditions such as diet, degree of exercise, infections, smoking, dust, pollution, and contact lens wear. Internal factors include an individual’s gut microbiome, metabolic factors, oxidative stress, genome, and surface microbiome [6]. Exposomes interact with an individuals’ genome through epigenetic modifications, affecting how genes are expressed without modifying the DNA sequence itself. The internal exposome that has gained recent attention for its impact on OSDs is the ocular surface microbiome. The microbiome refers to the genetic make-up of the microbial communities that colonize specific tissues [7]. In the current literature, there is increasing evidence that the host ocular surface microbiome may play important roles in the immunomodulation of ocular surface components and the immunopathogenesis of OSDs [7]. Understanding changes in the ocular surface microbiome may help to establish the causative factors underlying ocular surface inflammation and scarring, the hallmarks of OSDs. This review aims to provide an overview of the current evidence on the impact of internal and external exposomes, the ocular surface microbiome in particular, on various OSDs.

## 2. External Exposomes and the Ocular Surface

The ocular surface is directly exposed to a large array of external stimuli ranging from dust, pollution, weather, and temperature of the external environment to contact lens wear [4,5]. The epithelium of the entire ocular surface works as an immunological unit, and is important in the immunological defence against external insults. Ocular surface epithelial cells express specific immune pattern recognition receptors that can activate downstream innate and adaptive immunological cascades [2]. The triggering of receptors including Toll- and NOD- like receptors leads to the upregulation of transcription factors such as NF-κB and MAPKs, which in turn release inflammatory cytokines (e.g., IL-1, IL-17, TNF-α) and chemokine ligands, leading to both innate and adaptive immune responses [8], as shown in Figure 1. As a result, the interaction of external exposomes with the ocular surface can modulate inflammatory regulators such as cytokines, stress-response, hormones, and growth factors [9], and can initiate epigenetic mechanisms that may induce vulnerability of the ocular surface, resulting in an OSD. This paper serves to provide a current overview of the evidence on the impact of exposomes on OSDs and the proposed mechanisms of their action.

## 3. Internal Exposomes and the Ocular Surface

In OSDs, the ocular microbiome is the most extensively studied internal exposome. As mentioned above, interactions between ‘healthy’ commensal micro-organisms and epithelial cells’ receptors under physiological conditions account for immune homeostasis and tolerance. Other important factors reviewed here include oxidative stress and gut microflora.

### 3.1. The Ocular Surface Microbiome in Health

In health, the microbiome on the ocular surface is colonised by commensal microbes that do not lead to disease [7]. This community of micro-organisms, or microbiota, appears to be important in ocular surface immunoregulation [10,11,12,13]. In vitro studies have demonstrated that healthy cultured corneal and conjunctival epithelial cells do not mount an inflammatory response to known ‘physiological’ commensal bacterial organisms such as *Propionibacteria acnes* and *Staphylococcus epidermidis* [10]. On the contrary, these similar cultured cells express pro-inflammatory cytokines (e.g., IL6, IL8) when presented with known pathogens, such as *Pseudomonas aeruginosa* [10], and commensals such as Achromobacter [14]. Other investigators have also shown that, compared to wild type mice, germ-free mice not colonized with commensal bacterial organisms were predisposed to more severe *Pseudomonas* keratitis [11]. Similarly, specific-pathogen-free mice colonised with ocular *Corynebacterium* spp. demonstrated protective immune effects against *Pseudomonas* and *Candida* infections, with the ability to mount a stronger ocular immune T-cell cytokine response compared to non-colonised specific-pathogen-free mice [13]. These observations indicate that the composition and diversity of the ocular surface microbiota play an important role in the regulation of ocular immune responses. Defining the constituents of a ‘healthy’ ocular surface microbiome is challenging, however. Various factors, including age, environment, diet, and geographical location, can alter the ocular surface microbiome [15,16,17,18,19,20]. In general, studies of healthy subjects with normal conjunctiva have observed that three groups of bacterial phyla dominate the ocular surface microbiome: *Actinobacteria*, *Proteobacteria*, and *Firmicutes* [17,21,22]. At a genus level, the following micro-organisms dominate the healthy ocular surface microbiome: *Corynebacterium* spp., *Streptococcus* spp., *Propionibacterium* spp., *Bacillus* spp., *Staphylococcus* spp., and *Ralsontia* spp. Bacteria that are less-consistently identified in healthy eyes include *Pseudomonas* spp., *Escherichia* spp., *Actinetobacter* spp., *Acidovorax* spp., *Brevundomonas* spp., *Aquabacterium* spp., *Sphingomonas* spp., *Bradyrhizobium* spp., *Anaerococcus* spp., and *Ochrobactrium* spp. [17,21,22,23,24].

### 3.2. Alterations in Ocular Surface Microbiome in Disease

With the advancements in metagenomics techniques, alterations in the ocular surface microbiomes of patients with OSDs have been widely reported in recent years [25,26,27,28,29,30,31,32,33,34,35,36] (Table 1). Such changes in ocular surface microbiome constituents adversely modify the interactions with oral mucosa T-cells and interleukin 17 levels, compromising the local host defence against pathogens [13]. Pathogenic bacteria growth also results in lipases and toxins which damage the ocular surface and trigger an immune cascade of inflammation which, if uncontrolled, leads to further tissue damage in OSDs [27,37,38]. Understanding the specific changes in ocular surface microbiota and their interactions with various immunological components of the ocular surface will allow us to better understand the immunopathogenic mechanisms underlying OSDs.

## 4. Dry Eye Disease and Blepharitis/Meibomian Gland Dysfunction

Dry eye disease (DED) is very common, with some populations reporting a prevalence of symptomatic DED as high as 32.1% [39]. With a recently published international consensus on DED, there has been an increasing awareness of the diagnosis of DED over the past three decades [40]. Broadly speaking, DED can be classified into *evaporative* and *aqueous deficient* DED, with blepharitis/meibomian gland dysfunction (MGD) and lacrimal diseases (Sjögren/non-Sjögren) being the predominant underlying causes, respectively [41]. Tear hyper-osmolarity is thought to be the hallmark of DED, resulting in damage to ocular surface epithelial cells and the triggering of ocular surface inflammatory cascades, which perpetuates DED through a vicious cycle [41].

### 4.1. Impact of Exposomes on Dry Eye Disease

In addition to the ocular microbiome being an important catalyst for ocular surface diseases, the impact of exposomes on DED has been gaining interest [42]. A large variety of exposomes have been proposed as contributing factors to the pathogenesis and severity of DED. These range from external factors such as the humidity, temperature and airflow of the environment, urban or rural areas of living, air pollution, contact lens use, infections, use of visual display units, and illumination, to internal factors such as an individual’s gut microbiome, metabolic factors, oxidative stress, and inflammation [6,43].

### 4.2. External Exposome on Dry Eye Disease and Meibomian Gland Dysfunction

#### 4.2.1. Environmental Pollution

A review on the impact of environmental pollution on dry eye disease identified that multiple factors in both outdoor and indoor environments exert a significant impact on the incidence of dry eye disease and meibomitis [44]. Nitrogen oxide (NO_2_), carbon monoxide (CO), and particulate matter less than 10 microns in size (PM10) are the main compounds implicated in dry eye disease and meibomitis in studies of outdoor environments. A study of 55 healthy individuals in Sao Paolo, for example, showed that higher levels of NO_2_, a pollutant typically found in urban areas with high traffic pollution, correlated with an increased frequency of meibomitis and shorter tear break up times (TBUT) [45]. Another study in Sao Paolo found increased meibomian gland discharge and eyelid debris with increased concentrations of combustion-derived pollutants from vehicle emissions, such as CO, PM10, and NO_2_ [46]. An even larger study in South Korea with 16,824 participants found that higher ozone levels, lower humidity, and higher NO_2_ levels were significantly associated with dry eye disease [47]. A survey amongst 298 Singaporeans affected by regional haze from Indonesian forest fires showed that a high proportion (60.7%) of these individuals experienced significant eye discomfort [48]. The abovementioned pollutants exert oxidative stress when deposited on the ocular surface, overloading the antioxidative defence mechanisms, modifying the chemical structure of antioxidants, and causing chronic inflammation [49]. NO_2_ in the air, in particular, induces globlet cell hyperplasia in human tarsal conjunctiva [50]. Oxidative damage by ozone molecules or volatile aromatic organic compounds (VOC) can lead to the activation of stress pathways such as NF-kB, which increases the production of inflammatory cytokines, reducing mucin-secreting cells and causing the corneal epithelial integrity to breakdown [51].

#### 4.2.2. Humidity, Temperature, Cleanliness and Screen Time

As for indoor environments, factors such as humidity, temperature, cleanliness, and occupational factors such as screen time have been identified as key players in the development of dry eyes and ocular surface symptoms. A study in Massachusetts of 98 individuals found that a lack of office cleanliness and floor dust were significantly correlated to the presence of ocular surface symptoms [52]. A study of 3335 employees in Japan showed that eye irritation correlated with carpeting, coldness and humidity perception, high mental workload, and the presence of dust and dirt [50]. The build-up of dust and reduced humidity, along with higher screen time resulting in reduced blink rate and increased interpalpebral aperture (and increased ocular surface exposed), contribute to reduced TBUT and an increase in dry eye symptoms [53].

#### 4.2.3. Contact Lens Wear

Contact lens wear has also been found to contribute to dry eye disease, as contact lenses separates the tear film into pre-lens and post-lens tear films, resulting in the thinning of both tear film thickness and instability of the pre-lens tear film, as well as increased friction between the lens and the ocular surface [54]. A study of 4393 office workers in Japan found that contact lens users were 3.61 times more likely to have severe dry eye symptoms than non-contact lens users [55]. This finding was corroborated by the epidemiology subcommittee of the international dry eye workshop [56], which identified contact lens wear as a consistent risk factor for DED. In addition, different cleaning methods of contact lenses have been shown to affect the resulting pH and osmolality of the lens [57], with impacts on wearer comfort and tear film stability [58].

### 4.3. Internal Exposomes on Dry Eye Disease and Meibomian Gland Dysfunction

#### 4.3.1. Ocular Surface Microbiome

Alterations in the ocular surface microbiome have also been studied in DED, and it is hypothesized that such changes may be important in the immunopathogenesis of DED.

One study investigating the ocular surface microbiome of patients with clinical DED found an abundance of ocular surface commensals, including *Coagulase-negative Staphylococcus*, *Staphylococcus epidermidis*, *Corynebacterium* sp., *Propionibacterium*, and *Bacillus* spp., compared to controls [27]. More importantly, the investigators also showed an increase in bacterial flora and a relatively higher abundance of pathogens, including *Rhodocossus* spp. and *Klebsiella* spp., in DED participants that was significantly correlated with a depletion of goblet cell densities. These findings indicate a possible causative mechanism for the development of DED.

Similarly, another group also found significant variations in the ocular surface microbiome components of participants with DED compared to those without DED [29]. In this study, the investigators reported significant variations in the ocular surface microbiome at both the phylum and genus levels between DED and non-DED subjects. Ten bacterial phyla dominated most of the sequences from both groups: *Proteobacteria, Firmicutes, Bacteroidetes*, *Actinobacteria*, *Cyanobacteria*, *Acidobacteria*, *Chloroflexi*, *Planctomycetes*, *Epsilonbacteraeota*, and *Verrucomicrobia*. Bacterial genera which were common to both groups included *Pseudomonas*, *Acinetobacter*, *Bacillus*, *Chryseobacterium*, and *Corynebacterium*. However, the ocular microbiomes in DED subjects were enriched with *Bacteroidia* and *Bacteroidetes*, suggesting that these micro-organisms may be important in the pathophysiology of DED. Conversely, there were also lower levels of *Pseudomonas* spp. and *Proteobacteria* in DED subjects compared to controls. It was also observed that the ocular surface community of non-DED participants exhibited significantly greater phylogenetic diversity and dominance compared to the DED participants.

Studies have also evaluated changes in the microbiome in patients with blepharitis and MGD. In the same study reported above, when subjects with DED were divided into those with and without MGD, the investigators did not observe a difference in the ocular surface bacterial diversity between the two groups [29]. Interestingly, *Bacillus* organisms were detected at a higher relative abundance in samples obtained from participants with MGD compared to those without MGD. On the contrary, patients without MGD had a higher abundance of *Bacteroidetes.* In another small study evaluating the ocular microbiome of subjects with blepharitis, a relative increase in the abundance of *Staphylococcus*, *Streptophyta*, *Corynebacterium*, and *Enhydrobacter*, and a relatively lowered abundance of *Propionibacterium*, were observed in these patients compared to healthy controls [28]. As *Streptophyta*, *Corynebacterium*, and *Enhydrobacter* are found in plant pollens, soil, and dusts, the investigators postulated that blepharitis might be induced by infestations of micro-organisms found in these environmental agents.

In a larger study comparing Chinese subjects with MGD to healthy sex- and age-matched subjects, investigators reported a significantly higher abundance of phyla *Firmicutes* and *Proteobacteria* in subjects with MGD compared to controls; *Actinobacteria* was found to be lower in abundance [30]. At the genus level, *Staphylococcus* and *Sphingomonas* were significantly more abundant in the ocular surfaces of patients suffering from MGD compared to control eyes, whilst *Corynebacterium* was observed to be less abundant. In patients with MGD, the investigators also observed a direct correlation between *Staphyloccocus* abundance and their meibomian gland severity score. More recently, a study has further shown that meibum in MGD subjects contains distinctive microbiota when compared to healthy control subjects [31]. In this study, meibum obtained from MGD showed an abundance of *Campylobacter* spp. and *Enterococcus* spp. These pathogenic bacteria were not observed in controls. Through functional evaluations, the investigators demonstrated that the micro-organisms in MGD samples expressed genes associated with chemotaxis, immune-evasive virulence, and mediators of type IV hypersensitivity reactions. Indeed, such alterations in the ocular surface microbiome may play a significant role in the underlying mechanisms of DED and blepharitis/MGDs. However, it still remains unclear whether these changes in ocular surface microbiota result in the direct activation of inflammatory cascades seen in DED or whether they unfavourably alter the ocular surface components, predisposing it to atypical colonisation by micro-organisms.

#### 4.3.2. Gut Microflora, Laryngopharyngeal Reflux (LPR) Disease

The presence of gastro-oesophageal reflux disease (GERD) has been found, in recent studies, to be associated with dry eye disease [59,60,61,62,63]. The current potential mechanisms explaining this association are an alteration in nasolacrimal duct microbiota by the reflux episodes [64], or chronic inflammation and fibrosis caused by the backflow of LPR (the most common extra-oesophageal manifestation of GERD) into the nasolacrimal duct. Studies have found that, in patients with dry eye disease, there was a local increase of eye pepsin concentration [59,63], which is postulated to have travelled into the lacrimal system via the nasal cavity, inferior meatus, or nasolacrimal duct, and this may affect ocular surface though its direct proteolytic activity and the local expression of proinflammatory cytokines [65]. A study of 50 patients with LPR, proven both endoscopically and via symptom indices, found that tear pepsin levels correlated significantly with the severity of the LPR disease and ocular surface changes [66]. Pepsin, a serine-protease, can cause direct erosion of the mucosa and elicit hyperaemia and irritative symptoms, while at the same time inducing the production of pro-inflammatory cytokines; hence, it is postulated to similarly promote inflammatory cytokines when present on the ocular surface [61,66]. In addition, pepsin is a mucolytic [67] and, hence, has been postulated to have the ability to impair the ocular mucus layer, cause tear film disruption, and worsen ocular surface disease [61]. Helicobacter pylori also has a similar mechanism of reaching the lacrimal system, and is also associated with the release of proinflammatory and vasoactive molecules such as tumour necrosis factor-alpha, interleukins, interferon gamma, and prostaglandins that would also lead to mucosa damage and chronic inflammation [65].

#### 4.3.3. Impact of Exposome and the Drive towards Chronic Inflammation in Dry Eye Disease

Left unchecked, the impact of external and internal exposomes serves to continuously stimulate the corneal nerves—from factors such as increased evaporation from reduced humidity, or the presence of VOC causing oxidative stress and pollutants causing ocular surface toxicity. This results in the constant stimulation of the ocular surface’s adaptive immune responses, followed by a restarting of the innate immune response thereafter when the inciting triggers are not addressed adequately [68], resulting in a vicious cycle of dysregulation of the innate and adaptive phases and driving patients toward chronic ocular surface inflammation and disease. Moving forward, it is imperative to raise awareness of exposomes as significant contributors towards the severity and continued activity of dry eye disease, so as to garner more evidence on the importance and effectiveness of addressing exposomes together with treatment of dry eye disease in order to change the disease course of these patients and reduce the dry eye disease burden.

## 5. Allergic Eye Disease

A wide spectrum of allergic eye diseases affects the ocular surface, each with a different immunopathogenesis. Broadly speaking, allergic eye diseases can be divided into allergic conjunctivitis and two sight-threatening forms, namely atopic keratoconjunctivitis and vernal keratoconjunctivitis [69]. Allergic conjunctivitis can be further divided into perennial allergic conjunctivitis (PAC) and seasonal allergic conjunctivitis (SAC). Type I hypersensitivity is the hall mark of allergic conjunctivitis, where allergen-induced IgE cross-linking results in mast cell degranulation and the triggering of acute inflammation that includes a release of histamine [69]. In atopic keratoconjunctivitis (AKC) and vernal keratoconjunctivitis (VKC), in addition to Type I hypersensitivity triggered by allergens, there is also the activation of cell-mediated Type IV hypersensitivity. In VKC, there appears to be a recruitment of T-helper type 2 (Th2) lymphocytes, which are involved in the antibody response through B-lymphocytes, attracting more eosinophils that themselves may stimulate B-lymphocytes [70]. AKC is the most severe form of allergic eye diseases. In addition to Th2 T-lymphocytes, there is also T-cytotoxic type 1 (Th1) lymphocyte activation, which drives cell-mediated immunopathogenesis and macrophage-mediated tissue damage [70]. In both VKC and AKC, ocular morbidity results from chronic inflammation and ocular surface scarring, the latter occurring through various pathways including IL-13 and TGF ß-induced fibroblast activation [71].

The prevalence of allergic diseases, both systemic [72,73,74] and ocular [75,76], appears to be rising worldwide. Various hypotheses for this observation have been suggested. Amongst these area lowered exposure to environmental micro-organisms when the immune system is immature and an altered microbiome [77,78].

### 5.1. External Exposomes and Allergic Eye Disease

#### 5.1.1. Contact Lens Wear, Pollution, Lack of Exposure to Microbes

External exposome has been implicated in the increased prevalence of allergic diseases. Chief among the multitude of external factors is the reduced exposure to microbes due to our improved standards of living, which is theorised to impede the development and training of the immune system to develop tolerogenic responses, as well as the increased exposure to pollutants in our environment [79]. The epithelial-mesenchymal trophic unit (EMTU) is important for maintaining homeostasis and facilitating the repair of ocular tissues [80]. Dysregulation of the EMTU by exposomes such as pollutants, chemical injury, and trauma can lead to increased autophagy markers such as LC3B, Cathepsin D, Beclin-1, and LAMP1 in disease states, such as in VKC [81], resulting in barrier dysfunction via loss in major tight junctions and adhesion proteins, propagating inflammation and tissue remodelling. This further perpetuates the vicious cycle of ocular allergies [6]. Specifically, significant levels of ozone in the environment have been shown to result in the overexpression of IL-6 and tumour necrosis factor-alpha, which results in allergic ocular signs such as chemosis, reduced TBUT, ocular surface staining, and conjunctival injection [82]. Ozone particles and nitrous oxide have been shown to cause direct damage to ocular mucosa and induce goblet cell hyperplasia in the conjunctiva [50] via their high oxidative potential. Other pollutants such as sulfur dioxide, nitrous oxide, and contact lens wear have been shown to lower the pH of tears [83], causing irritation of the ocular surface [84] and enhancing the allergic sensitization of ocular tissues [85]. Diesel exhaust particle exposure also led to an increase in the expression of pro-inflammatory cytokines and chemokines such as intercellular adhesion molecule 1 and interleukin 6 [86].

A large retrospective study on 15,938,870 patients with over 3,211,820 visits for allergic conjunctivitis found a correlation of levels of nitric oxide, ozone, and temperature with the number of visits for allergic conjunctivitis [87], hence supporting the possibility of ambient air pollution and weather worsening allergic conjunctivitis. Previous studies have also supported the impact of exposomes on allergic eye disease, with a study of 15 subjects showing a correlation of symptoms of rhinoconjuntival tissue irritation with ambient air pollution levels [88]. A Japanese study of 3004 individuals also found that the prevalence of the severe forms of allergic conjunctivitis such as AKC and VKC were significantly associated with the levels of the air pollutants, specifically NO2 for AKC and NOx and PM10 for VKC, respectively [89]. From these studies, it is apparent that the exposome plays an important role in the pathogenesis and propagation of allergic eye diseases.

#### 5.1.2. Diet

Diet has also been shown to have an impact on the severity of symptoms of allergic eye disease. In an analysis of data collected by the International Study of Asthma and Allergies in Childhood (ISAAC) programme of 721,601 children across 56 countries, an association was found between the regression of symptoms of allergic rhinoconjunctivitis and an increased per capita consumption of cereal, rice, and nuts, as well as vegetables. This was postulated to be possibly contributed to by the antioxidant effects of vitamin A and E, found in these food sources, exerting a protective function against symptoms of allergic rhinoconjunctivitis, while other potential links have yet to be identified [90].

### 5.2. Internal Exposomes and Allergic Eye Disease

#### Ocular Surface Microbiome

Alterations in the ocular surface microbiome in patients with allergic eye diseases are not well reported in the current literature. Using metagenomics shotgun sequencing, one study which evaluated 32 patients with allergic eye diseases (21 SAC/PAC and 18 VKC) showed that the conjunctival microbiome in these patients was distinct to the microbiome of healthy control subjects [32]. Bacteria dominated the ocular surface microbiome of all participants, with a lower inter-individual variation in alpha diversity of the allergic eye disease participants compared to healthy controls. Interestingly, *Malassezia* fungi were found to be abundant in a fraction of patients with SAC/PAC. The authors postulated that, as *Malassezia* is known to produce antigenic proteins that can trigger Ig E-mediated immunogenic responses in atopic skin diseases [91,92], this alteration in the microbiome may be significant in the pathophysiology of SAC/PAC. Furthermore, the authors found an enrichment of *Moraxella catarrhalis* in patients with allergic eye diseases. Being a known important contributory factor in the exacerbation of allergic respiratory disease [93], such an abundance of *Moraxella* spp. in the ocular surface microbiome may indicate a comparable pathophysiology in both systemic allergies and allergic eye diseases. Lastly, when conjunctival samples obtained from patients with SAC/PAC were compared to those from VKC patients, the investigators reported a significant variation in the microbiome between the groups. In particular, they observed an increase in the relative abundance of *Brevibacterium* spp., *Staphylococcus* spp., *Hymenobacter* spp., and *Microbacterium* sp. in samples obtained from SAC/PAC patients. In contrast, there is an increase in relative abundance of *Streptococcus* spp., *Auricoccus* sp., *Prevotella* sp., *Actinomyces* sp., and *Campylobacteri* sp. in VKC patients. These findings highlight the differences in microbiome compositions in different forms of allergic eye disease, which represent the different underlying disease mechanisms, resulting in the distinct clinical phenotypes.

Another study investigated 22 children with VKC compared to healthy age-, sex-, and ethnicity-matched controls using high throughput 16S rRNA sequencing [33]. Similarly, this group of investigators found a higher abundance of *Moraxella* sp. in the ocular surface of VKC subjects compared to healthy controls at the phylum level. In addition to *Proteobacteria*, *Firmicutes*, and *Actinobacteria*, which were found in the core microbiomes of all participants in this study, *Bacteroidetes* and *Fusobacteria* were also found in the samples obtained from VKC patients. The authors hypothesized that such alterations in the ocular surface microbiome with the additional presence of gram-negative bacteria in VKC subjects may potentially induce a lipopolysaccharide (LPS)-induced inflammatory response, suggesting a molecular mechanism for VKC [94]. When the investigators evaluated the fungal microbiome, *Malasseziaceae* was observed to be significantly greater in abundance in patients with VKC compared to controls [33]. Through conjunctival RNA sequencing transcriptomics, this group of investigators have previously shown an over-expression of pattern recognition receptors in VKC [95]. Thus, they hypothesized that *Malasseziacea* interacts with these receptors, triggering a Th2-like response that is similar to that seen in atopic skin diseases [92].

## 6. Cicatrising Conjnctivitis

Cicatrising conjunctivitis (CC) is a heterogenous group of sight-threatening diseases with characteristic hall marks of ocular surface inflammation and scarring [96,97,98]. In developed countries where trachoma, an important world-wide cause of CC, has been eliminated, Stevens-Johnson syndrome (SJS) and mucous membrane pemphigoid (MMP) with ocular involvement are the most common causes of CC [99].

### 6.1. External Exposomes on Cicatrising Conjunctivitis

#### Viral Infections and Drugs

Similar to the impact of exposome on that of allergic eye disease, external exposomes can result, again, in EMTU dysregulation and the loss of tolerance to one or more components of the basal membrane zone [100], resulting in ocular surface remodelling, such as that of progressive shortening and subepithelial fibrosis in mucosal membrane pemphigoid [6]. In SJS, viral infections and environmental triggers such as drugs can activate the toll-like receptors, activating the innate immune system and mediating the production of pro-inflammatory cytokines, resulting in damage of the ocular surface [101]. However, specific studies on the impact of specific exposomes on SJS and MMP are still lacking, and the exact triggers for the onset of ocular MMP are still unknown, partly due to the multifactorial nature of the exposomal environment, as well as the relative rarity of such patients.

### 6.2. Internal Exposomes on Cicatrising Conjunctivitis

#### 6.2.1. Gut Microbiome

In addition to having an impact on dry eye disease, the gut microbiome may play a role in the pathogensis of cicatrising conjunctivitis as well. A case report of a patient with ulcerative colitis (UC) and concomitant MMP reported remission of his ocular disease after a colectomy [102]. The authors of the case report postulate that UC resulted in increased translocation of the gut microbiome, and the resulting increased antigenic activity had cross reactivity with the basement membrane zone proteins of ocular tissues, which was also previously supported in a case series on six patients with UC and immunobullous skin disease, in which the temporal sequence of UC followed by subsequent development of the skin disease strongly suggested that the bowel inflammation initiated the immune response to cutaneous antigens [103]. Further studies on other external exposomes’ impact on patients with SJS and MMP would also allow us to better understand the pathways driving these disease processes and would potentially aid in prognostication and management strategies.

#### 6.2.2. Ocular Surface Microbiome

Similar to other OSDs, alterations in the ocular surface microbiota have been previously reported through traditional culture techniques [35,104,105]. Investigators showed that gram-positive bacteria, namely *Staphylococcus* spp. And *Corynebacterium* spp., were more frequently isolated from the conjunctiva of patients with ocular SJS compared to healthy controls. The atypical organisms *Serratia* spp., *Escherichia coli*, and *Proteus mirabilis*, *Haemophilus* spp. have also been reported to colonise the ocular surface of these patients [35,105].

More recently, the ocular surface microbiome of ocular SJS patients using metagenomics sequencing techniques have also been reported. The first of these studies was a small case series where investigators reported a higher proportion of *Staphylococcus* in the ocular surface of SJS patients compared to controls [34].

In this study, higher levels of *Corynebacterium* were also seen in several SJS patients. Other observed differential colonisations between SJS patients and controls include *Lactobacillus*, *Prevotella*, *Fusobacterium*, and *Enterobacteriaceae*.

In another study comparing 20 patients with chronic ocular SJS to 20 healthy control participants, the investigators showed a significant variation in the core microbiome between the two groups [35]. In addition to the *Pseudoalteromonadaceae* and *Vibrionaceae* families which were found in both groups, *Burkholderiaceae* and *Enterobacteriaeceae* were also found in the ocular SJS group. Moreover, at a genus level, there was a greater abundance of *Acinetobacter* spp., *Bacteroides* spp., *Faecalibacterium* spp., *Prevotella* spp., *Corynebacterium* spp., *Pseudomonas* spp., *Staphylococcus* spp., and *Streptococcus* spp. in the SJS group. In contrast, a higher abundance of *Vibrio* spp., *Acrobacter* spp., *Clostridium* spp. *Cetobacterium* spp., and *Fusibacter* spp. were found in the healthy control group. Interestingly, whilst culture techniques have found that gram-positive bacteria are the most common isolated colonisers of the ocular surface in SJS patients, this metagenomics study revealed not only a larger diversity of bacterial community in the ocular surface microbiome, but gram-negative bacteria appeared to dominate. These findings indicate the higher sensitivities of metagenomics analyses using high throughput sequencing to detect pathogenic micro-organisms which may be slow growing and difficult to culture.

Using a similar metagenomics technique, another study comprising of 37 SJS patients with severe ocular complications and 9 healthy control subjects, conducted in Japan, reported a significant reduction in bacterial diversity in the SJS group [36]. The ocular surface microbiome between the SJS and control participants were also significantly different. At a genus level, there was enrichment of *Corynebacterium* spp., *Neisseriaceae* spp., *Staphylococcus* spp., *Propionibacterium* spp., *Streptococcus* spp., *Escherichia* spp., *Fusobacterium* spp., *Lawsonella* spp., and *Serratia* in the SJS group compared to controls. Temporal stability was also demonstrated, with no change in the microbiome of participants seen over two separate sampling timepoints. These alterations in the microbiome found in patients with ocular SJS, who also had an increased colonisation of pathogenic micro-organisms, may explain the increased risk of severe blinding infections seen in these eyes.

Some postulated mechanisms through which the microbiome contributes to the pathogenesis of SJS occur through the compromise of the immunosuppressive environment of the eye and its innate immunity response, upsetting the balance of mucosal immunity and pathogenicity of the surface microbiome and resulting in a chronic and recurrent ocular surface inflammation [10,104,106]. Commensals such as coagulase negative staphylococcus, when identified predominantly from the conjunctiva swabs of SJS patients, seem to be accompanied with severe ocular surface abnormalities such as chronic corneal epitheliopathy and a reduced mucin layer of the tear film [35]. Pathogenic bacteria, especially in the context of altered conjunctival immunity due to abnormal eyelid structures in SJS, such as entropion and trichiasis, conjunctival scarring, and corneal changes, also result in a higher incidence of opportunistic infections [104,107]. However, studies on the temporal sequence of events, i.e., whether the identified microbiome of SJS patients triggered alterations in ocular innate immunity responses or the reverse is uncertain. Understanding the cause and effect of these findings would help in the management of OSD.

## 7. Cosmetics: Common but Often Overlooked External Exposome with Impact on the Ocular Surface

Ocular cosmetic use is increasingly widespread and consists of an extensive range of leave-on and wash-off products [108]. These products contain a myriad of cosmetic ingredients which function as abrasive, absorbent, buffer, colourant, pH adjuster, or surfactant [108]. As reported comprehensively in the Tear Film and Ocular Surface (TFOS) Lifestyle report on the impact of cosmetics on the ocular surface [108], due to the thin eyelid and periorbital skin, compounds in eye makeup and skincare products such as retinoids [109] and tea tree oil [110] can easily penetrate and migrate onto the ocular surface, causing negative effects such as orifice obstruction of the meibomian glands, promoting ocular surface inflammation and damage, and worsening meibomian gland and dry eye disease [111,112,113,114]. In addition, numerous substances in products such as eyeshadows, mascara, eyeliner, and eye creams, for instance, benzalkonium chloride [115], parabens, phenoxyethanol [116], shellac, and 1,3-butylene glycol, have been shown to be toxic to the ocular surface and cause MGD and eyelid contact dermatitis [117,118,119]. A cross sectional study of 42 healthy women also found that tear breakup time was significantly lower in the eyeliner use group as compared to the non-eyeliner use group, with a higher incidence of MGD and conjunctival inflammation [120]. Common ingredients in skincare products, such as ceramides and free cholesterols, have also been shown to disrupt meibum stability, resulting in an unstable tear film due to increases in hysteresis, rigidity, and collapsibility of the resulting mixture of meibum and ceramides or free cholesterols, as shown in Arciniega et al. [121]. Studies attempting to study the tear lipid–aqueous interface via the use of simplified models have also found that lipid composition, including that of polar lipid biomimetics [122,123] and ceramides [124], may alter tear film stabilisation depending on their composition after multiple compression-decompression cycles.

Cosmetic procedures around the eyes such as eyelash extensions, tattoos, and injections are also increasingly popular. Eyelash extensions, however, are associated with allergic contact dermatitis and blepharitis [125]. Periocular Botulinum injections [126,127] and eye lid tattoos [128,129] can impair meibomian gland secretions and result in tear film instability.

In addition, cosmetic products such as makeup brushes and sponges serve as reservoirs for microbial growth. A study of samples from 100 brushes and sponges found Staphylococcus aureus in all of them, Pseudomonas aeruginosa in 81.8% of brushes and 69.6% of sponges, and fungus in 30.3% of brushes and 51.5% of sponges [130]. Considering the ubiquitous use of these products, further studies on how makeup use might affect the ocular surface microbiome as well as how long-term exposure to these products might adversely affect ocular surface health would be immensely valuable for the development of future safety guidelines for the production of periocular products and tools, and agents used in cosmetic procedures around the eye.

## 8. Contact Lens Wear: An Example of External Exposome Impacting upon Internal Exposome of the Ocular Surface

Contact lens wear is a unique environmental factor in that it is an external exposome by definition, contributing to ocular surface diseases such as allergic eye disease and DED. However, contact lens wear also significantly alters the inner exposome of the patient, i.e., the surface microbiome. Contact lens wear is also one of the most important risk factors for all forms of corneal infections (infectious keratitis) in developed nations, accounting for over 60% of diagnosed cases [131,132]. The causative organisms for such infections are mostly bacterial, with *Pseudomonas* spp. being the most-commonly isolated [131,133]. Fungal, amoebic, and other atypical pathogens, although less commonly encountered in clinical practice, are increasingly being reported as important emerging organisms in contact lens-related infectious keratitis [132,134].

Due to the challenges in the identification of causative organisms resulting in delays to initiating appropriate therapies, infections caused by such atypical pathogens are often associated with more unfavourable clinical outcomes [134]. Thus, identifying underlying factors that lead to these sight-threatening corneal infections is important.

Changes in the ocular surface epithelium and the microbiological community in subjects who wear contact lenses have been widely reported; these changes are thought to be the driving factors that lead to corneal infections [21,25,135,136,137,138,139]. The differences in ocular surface microbiota in contact lens wearers compared to non-contact lens wearers were first reported in studies using traditional culture techniques [136,137]. One such study showed that, compared to controls, daily wearing of soft contact lenses increases the number of lid and conjunctival commensal non-pathogenic bacteria that can be isolated [137]. More pathogens can be isolated from the ocular surface of extended-wear soft contact lens wearers [137]. Another study also showed the consistently greater overgrowth of gram-positive bacteria (*Coagulase-negative staphylococcus, Propionibacteria* sp., *Bacillus* spp., *Streptococcus* spp., *Micrococcus* spp., *Staphylococcus* spp., *Corynebacterium* spp.) in contact lens-wearing children [138]. This was more significant in samples obtained from the lower lid margins compared to the upper lid margins [138]. Another study reported that the conjunctiva tended to be significantly colonised with bacteria after using continuous-wear silicone hydrogel contact lenses; the study showed an increase in the number of eyes’ culturing of *Coagulase-negative staphylococci* and *diphtheroid rods* as determined by conjunctival sampling [139]. Thus, these findings indicated that contact lens induced changes on the ocular surface microbiota appear to depend on the type and duration of contact lens wear.

Interestingly, the type of contact lens solution and cleaning regime has been implicated in the alteration of the ocular microbiome. Retuerto et al. found that the abundance of Corynebacterium, Haemophilus, and Streptococcus was increased 4.3-, 12.3-, and 2.7-fold, respectively, in lenses cleaned with multipurpose solutions as compared to hydrogen peroxide solutions [140].

Furthermore, alterations in ocular surface microbiota in contact lens wearers have been implicated in contact lens-related diseases. For example, investigators have reported that, in contact lens wearers, gram-positive bacteria (e.g., *Coagulase-negative Staphylococcus* or *Corynebacterium* spp.) isolated from contact lenses or their ocular surface were more likely to develop contact lens-associated corneal infiltrates [141]. In addition to corneal infiltrates, contact lens wearers in who gram-negative bacteria has been isolated (e.g., *Haemophilus* spp.) is a risk factor for the development of contact lens-associated acute conjunctival hyperaemia [141,142].

Furthermore, one group of investigators more recently showed that the conjunctiva of individuals who wore contact lenses had bacterial community structures more akin to those observed in skin [25]. This group observed higher abundances of *Methylobacterium*, *Lactobacillus*, *Acinetobacter*, and *Pseudomonas*, and lower abundances of *Haemophilus*, *Streptococcus*, *Staphylococcus*, and *Corynebacterium* compared to non-contact lens wearers [25]. The authors postulated that such observed changes in the microbiome of contact lens wearers may be the result of the direct transfer of skin bacteria (from hands or eyelid) to the ocular surface or contact lenses exerting differential loads on the ocular surface microbiota in favour of skin organisms [25]. Using a similar metagenomics technique, another group also observed changes in the relative abundance of bacteria in the ocular surface microbiome [26]. This group found a lower abundance of *Bacillus*, *Lactobacillus,* and *Tatumella* in subjects who wore orthokeratology lenses compared to non-contact lens wearers. Evaluating those who wore soft contact lenses showed a lower abundance of *Delftia*, whilst the abundance of *Elizabethkingia* increased [26]. Nevertheless, whether such alterations in the ocular surface microbiome in contact lens wearers affect the ocular surface defence mechanisms and provide less protection from corneal infections still requires further evaluation.

## 9. Conclusions

This review has demonstrated how exposomes have a multifactorial and variable contribution to the pathogenesis and exacerbation of various OSDs. In particular, alterations in the ocular surface microbiome are common in various OSDs. Variations in the micro-organism communities have also been reported in different disease phenotypes, such as that seen in allergic eye diseases. However, it is unclear if these changes in the microbiota result in the pathogenesis of diseases or are the sequelae of the diseases. Indeed, the interactions between micro-organisms and components of the ocular surface immune system are complex and still remain to be elucidated. It is only through a deeper understanding of such relationships between exposomes, both internal exposomes, such as ocular surface microbes, and external exposomes, such as pollution, that potential alternative therapies can be developed. However, detailed evidence of the impact of specific exposomes on OSDs is often difficult to attain, as it requires the accurate collection of multiple concomitant external stimuli that patients are exposed to, while at the same time recording internal environmental responses. The dynamic nature of exposomes also makes studying of the exposome tricky, as parameters are changing continuously. The impact of exposomes on the increasing disease burden, however, necessitates further research in this area, as greater knowledge of significant exposomal factors promises a change in the prognostication and management of OSDs.

## Figures and Tables

**Figure 1 ijms-24-11273-f001:**
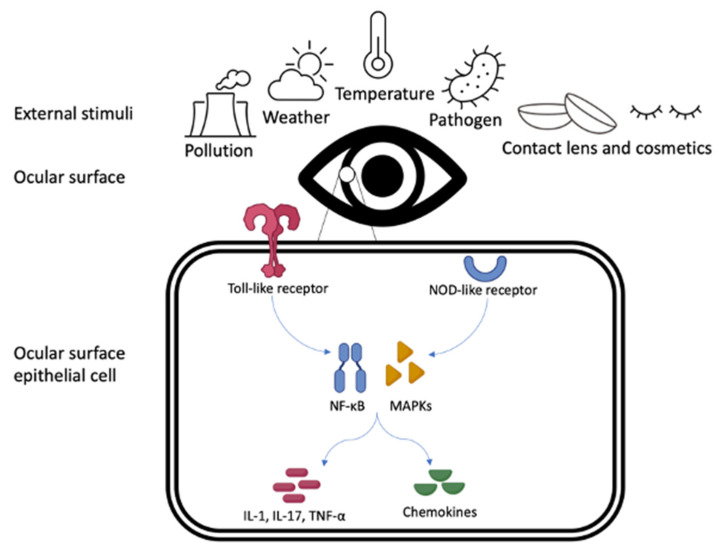
Representation of the Toll- and NOD-like receptor pathways within the ocular surface epithelial cell, in response to external stimuli.

**Table 1 ijms-24-11273-t001:** Representation of ocular surface microbiome, detected by metagenomics methodologies in various ocular surface diseases.

Ocular Surface	Studies	Study Population (Eyes)	Genus-Level Ocular Surface Microbiome
Disease	(Year)		
Contact lens wear	Shin et al. [23] (2016)	9 contact lens wearers	Higher abundance *:
		11 non-contact lens wearers	*Methylobacterium*
			*Lactobacillus*
			*Acinetobacter*
			*Pseudomonas*

			Lower abundance *:
			*Haemophilus*
			*Streptococcus*
			*Staphylococcus*
			*Corynebacterium*
	Zhang et al. [24] (2017)	20 OKL wearers	OKL wearers
		22 SCL wearers	Lower abundance *: *Bacillus, Tatumella, Lactobacillus*
		25 non-contact lens wearers	

			SCL wearers
			Higher abundance *: *Elizabethkingia*
			Lower abundance *: *Delftia*
Dry Eye Disease	Graham et al. [25] (2007)	57 Non-dry eye disease	Higher abundance *:
		34 Dry eye disease	*Coagulase negative staphylococcus*
			*Staphylococcus*
			*Corynebacterium*
			*Propionibacterium*
			*Bacillus*
	Lee et al. [26] (2012)	7 Blepharitis	Higher abundance *:
		4 Healthy controls	*Staphylococcus*
			*Streptophyta*
			*Corynebacterium*
			*Enhydrobacter*

			Lower abundance *:
			*Propionibacterium*
	Li et al. [27] (2019)	54 Non-dry eye disease	Dry eye disease
		35 Dry eye disease	Higher abundance *:
		(25 MGD; 10 non-MGD)	*Bacteroidia*
			*Bacteroidetes*

			Lower abundance *:
			*Pseudomonas*
			*Protebacteria*

			MGD
			Higher abundance ^†^:
			Bacillus

			Lower abundance ^†^:
			Bacteroidetes
	Dong et al. [28] (2019)	47 MGD	Higher abundance *:
		42 Healthy controls	*Staphylococcus*
			*Sphingomonas*

			Lower abundance *:
			*Corynebacterium*
	Zhao et al. [29] (2020)	61 MGD	Higher abundance *:
		15 Healthy controls	Rubrobacter
			Novibacillus
			Campylobacter
			Geobacillus
			Sphingomonas
			Corynebacterium
			Sphingobium
			Pedobacter
			Fictibacillus
			Enterococcus
Allergic Eye Diseases	Liang et al. [30] (2020)	21 SAC/PAC	SAC/PAC
		18 VKC	Higher abundance ^‡^: *Brevibacterium, Staphylococcus, Hymenobacter, Microbacterium*

			VKC
			Higher abundance ^‡^: *Streptococcus, Auricoccus, Actinomyces, Campylobacter, Prevotella, Paracoccus, Atopobium, Candida*
	Leonardi et al. [31] (2021)	VKC	Higher abundance *:
		Heathy controls	*Haemophilus*
			*Rothia*
			*Corynebacterium*
			*Prevotella*
			*Bacillus*
Cicatrising conjunctivitis	Zilliox et al. [32] (2020)	12 ocular SJS	Higher abundance *:
		6 healthy controls	*Staphylococcus*
			*Corynebacterium*
			*Streptococcus*
			*Lactobacillus*
			*Prevotella*
			*Fusobacterium*
			*Enterobacteriaceae*
	Kittipibul et al. [33] (2021)	20 ocular SJS	Higher abundance *:
		20 healthy controls	*Bacteroides*
			*Faecalibacterium*
			*Salinivibrio*
			*Akkermansia*
			*Prevotella*
			*Coprococcus*
			*Streptococcus*
			*Lactobacillus*
			*Fusobacterium*
			*Bifidobacterium*
			*Blautia*
			*Bacillus*
			*Phascolarctobacterium*
			*Paraprevotella*
			*Acinetobacter*
			*Ruminococcus*
			*Megamonas*
			*Odoribacter*
			*Staphylococcus*
			*Pseudoalteromonas*
			*Erwinia*
			*Pseudomonas*
			*Collinsella*
			*Methanobrevibacter*
			*Veillonella*
			*Thermomonas*
			*Roseburia*
			*Turicibacter*

			Lower abundance *:
			*Vibrio*
			*Acrobacter*
			*Cetobacterium*
			*Methylophaga*
			*Tenacibaculum*
			*Fusibacter*
			*Clostridium*
			*Cohaesibacter*
			*Shewanella*
	Ueta et al. [34] (2021)	37 ocular SJS	Higher abundance *:
		9 healthy controls	*Corynebacterium*
			*Neisseriaceae*
			*Staphylococcus*
			*Propionibacterium*
			*Streptococcus*
			*Escherichia*
			*Fusobacterium*
			*Lawsonella*
			*Serratia*

OKL = orthokeratology lens; SCL = soft contact lens; MGD = meibomina gland dysfunction; SAC = seasonal allergic conjunctivitis; PAC = perennial allergic conjunctivitis; VKC = vernal keratoconjunctivitis; SJS = Stevens-Johnson Syndrome; * disease compared to healthy controls; ^†^ dry eye disease patients with meibomian gland dysfunction compared to those without; ^‡^ SAC/PAC compared to VKC.

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
