# Peer review of "Impact of Exposomes on Ocular Surface Diseases"

_ijms, 2023, doi:10.3390/ijms241411273_

Round 1

Reviewer 1 Report

The manuscript is well designed and organized. However, the main idea on ocular surface diseases was focused on microbes. Therefore, the title needs to have "Microbes" or "interaction with microbiomes". 

This is the only comments on the manuscript. 

Slight modification of sentence may be required, but not necessarily. 

Author Response

Dear reviewer,  

Thank you for taking the time and effort to review the manuscript. Attached below is our response:

Point 1: The manuscript is well designed and organized. However, the main idea on ocular surface diseases was focused on microbes. Therefore, the title needs to have "Microbes" or "interaction with microbiomes". 

Response 1: The title has been modified to be "Impact of Exposomes and Microbiome on Ocular Surface Diseases". Thank you for taking the time and for your kind review.

Yours Sincerely,

Merrelynn Hong

Reviewer 2 Report

The present review paper showcases an extensive amount of carefully reviewed bibliographic material, accompanied by a meticulous search effort concerning the topic of eye exposomes and their impact on the ocular surface. The language used is highly precise, and the structure is easily comprehensible. I extend my congratulations to the authors for their commendable work. Moreover, I would like to humbly offer a few comments, primarily related to the bibliographic aspects, which I believe could potentially add further value to this study.

- Line 46: A space is required between "[3]." and "Nevertheless."

- Lines 51-53: Please include a reference (e.g., https://pubmed.ncbi.nlm.nih.gov/37062427/ and/or https://pubmed.ncbi.nlm.nih.gov/36572605/).

- Lines 60-63: A reference is needed. I recommend utilizing some of the information from Table 1.

- Lines 69-71: Please provide the references. I suggest reusing the two references mentioned above for simplicity (https://pubmed.ncbi.nlm.nih.gov/37062427/ and https://pubmed.ncbi.nlm.nih.gov/36572605/).

- 2. External Exposomes and the Ocular Surface: I believe this section could benefit from the inclusion of a concise schematic representation illustrating the reported pathway. Such a visual aid would facilitate a seamless reading experience for the audience.

- Lines 136-137: The prevalence data could be augmented with more recent literature (e.g., https://pubmed.ncbi.nlm.nih.gov/37003925/).

- Lines 108-196: Since this document focuses on microbial infections and related aspects on the ocular surface, a brief discussion on not only contact lenses but also the cleaning methods for disinfection could enhance and reinforce the presented ideas (e.g., https://pubmed.ncbi.nlm.nih.gov/35165235/ or https://pubmed.ncbi.nlm.nih.gov/21912303/).

- 7. Cosmetics: A Common but Often Overlooked External Exposome with an Impact on the Ocular Surface: This section could be enriched by incorporating the latest information recently reported (https://pubmed.ncbi.nlm.nih.gov/37061220/ and/or https://pubmed.ncbi.nlm.nih.gov/31724986/).

The present manuscript shows a right English performance

Author Response

Dear reviewer,  

Thank you for taking the time and effort to review the manuscript. Attached below are our responses:

Point 1: - Line 46: A space is required between "[3]." and "Nevertheless."

Response 1: A space has been added.

Point 2: - Lines 51-53: Please include a reference (e.g., https://pubmed.ncbi.nlm.nih.gov/37062427/and/or https://pubmed.ncbi.nlm.nih.gov/36572605/).

Response 2: We have added the references as you have suggested.

Point 3: - Lines 60-63: A reference is needed. I recommend utilizing some of the information from Table 1.

Response 3: A reference has been given.

Point 4: - Lines 69-71: Please provide the references. I suggest reusing the two references mentioned above for simplicity (https://pubmed.ncbi.nlm.nih.gov/37062427/ and https://pubmed.ncbi.nlm.nih.gov/36572605/).

Response 4: The two references have been added as per your suggestion.

Point 5: - 2. External Exposomes and the Ocular Surface: I believe this section could benefit from the inclusion of a concise schematic representation illustrating the reported pathway. Such a visual aid would facilitate a seamless reading experience for the audience.

Response 5: Thank you for the suggestion, a figure has been added in lines 84-86.

Point 6: - Lines 136-137: The prevalence data could be augmented with more recent literature (e.g., https://pubmed.ncbi.nlm.nih.gov/37003925/).

Response 6: Reference and quoted figure has been updated.

Point 7: - Lines 108-196: Since this document focuses on microbial infections and related aspects on the ocular surface, a brief discussion on not only contact lenses but also the cleaning methods for disinfection could enhance and reinforce the presented ideas (e.g., https://pubmed.ncbi.nlm.nih.gov/35165235/ or https://pubmed.ncbi.nlm.nih.gov/21912303/).

Response 7: We have added a discussion regarding the effects of contact lens cleaning methods on tear film stability from lines 197-199, microbiome from lines 550-554. 

Point 8: - 7. Cosmetics: A Common but Often Overlooked External Exposome with an Impact on the Ocular Surface: This section could be enriched by incorporating the latest information recently reported (https://pubmed.ncbi.nlm.nih.gov/37061220/ and/or https://pubmed.ncbi.nlm.nih.gov/31724986/).

Response 8: Thank you, we have incorporated the information from TFOS Lifestyle report on cosmetics and the above into lines 492 to 507.

Yours Sincerely,

Merrelynn Hong